# Self-Predictive Universal AI

**Elliot Catt, Jordi Grau Moya, Marcus Hutter, Matthew Aitchison**
**Tim Genewein, Gregoire Deletang, Kevin Li Wenliang, Joel Veness**
Google DeepMind
ecatt@google.com

## Abstract

Reinforcement Learning (RL) algorithms typically utilize learning and/or planning techniques to derive effective policies. Integrating both approaches has proven to be highly successful in addressing complex sequential decision-making challenges, as evidenced by algorithms such as AlphaZero and MuZero, which consolidate the planning process into a parametric search-policy. AIXI, the universal Bayes-optimal agent, leverages planning through comprehensive search as its primary means to find an optimal policy. Here we define an alternative universal Bayesian agent, which we call Self-AIXI, that on the contrary to AIXI, maximally exploits learning to obtain good policies. It does so by *self-predicting* its own stream of action data, which is generated, similarly to other TD(0) agents, by taking an action maximization step over the current on-policy (universal mixture-policy) Q-value estimates. We prove that Self-AIXI converges to AIXI, and inherits a series of properties like maximal Legg-Hutter intelligence and the self-optimizing property.

## 1 Introduction

Reinforcement Learning (RL) [1] algorithms exploit learning, planning [1], or their combination, to obtain good policies from experience. Pure learning consists of using real experience for improving a policy via a (parametric) model, possibly representing an explicit policy-model and/or the Q-values [2–6]. In a sense, learning stores the computational effort of policy-improvement into the parameters, which makes it a computationally efficient approach when needing to reuse the policy later on. In contrast, pure planning finds good policies via simulated experience using an environment model and a randomized (or exhaustive) search policy [1, 7, 8]. In the case of unknown or stochastic environments, one must re-plan after receiving a new observation, thus wasting all computational-effort from the previous step. This makes pure planning a wasteful approach. Using both, planning and learning, is a good way to improve performance and efficiency as demonstrated by modern high-performant RL algorithms such as MuZero [9–12]. These algorithms distill the planning effort back into the parametric search-policy by training it to predict the good actions obtained from planning. In a way, these agents are *self-predicting* their own policy-improvements. Although empirically successful and widely used, this distillation [13] or self-prediction [2] process is motivated in a purely heuristic way without much theoretical understanding on its optimality condition.

The AIXI agent [14, 15] is a theoretical universal Bayes-optimal agent obtained through pure planning without relying on distilling the search effort as described above. AIXI learns an environment model via a Solomonoff predictor [16, 17] and uses it for exhaustive (computationally intractable) planning. Thus, although it uses learning for the environment model, we say AIXI adopts a pure planning approach in the context of policy generation. Two desirable properties of Solomonoff prediction are universality—obtained by considering a huge hypothesis class containing all computable

---

[1] We use the terms planning and search interchangeably.

[2] Policy distillation usually refers to the process of amortizing one or several policies into another policy model. We view self-prediction as a type of distillation where a search-policy is consolidated into another model.

37th Conference on Neural Information Processing Systems (NeurIPS 2023).

environments—and fast convergence to the true environment statistics [18, 19]. Unfortunately, universality makes Solomonoff prediction incomputable, a property that AIXI also inherits. Nevertheless, the AIXI agent is considered as the most powerful agent and the gold-standard in decision-making under unknown general environments. An existing gap in the literature, however, is the lack of an alternative universal agent that maximally exploits learning and distillation, instead of the more wasteful planning approach used by AIXI. Understanding this distillation process from a theoretical standpoint can inspire taking new directions in approximating the AIXI agent as done in [20], which is based on a pure planning approach.

The central aim of this paper is to define an agent, which we name Self-AIXI, that maximally exploits *self-prediction* (or distillation) instead of planning, and to prove that this agent is optimal in the sense that converges to the gold-standard AIXI agent. We define self-prediction as the process of predicting the stream of action-data generated by the agent itself. Self-AIXI generates action-data in a similar fashion as other Temporal Difference TD(0) [1] algorithms in which there is only a single "planning" step when choosing an action i.e. $\arg\max_{a_t} \hat{Q}(h_t, a_t)$ for history $h_t$, action $a_t$ and current Q-value estimates $\hat{Q}$. The difference lies in that Self-AIXI does exact Bayesian inference on the policy space by holding a universal mixture $\zeta$ over policies (in addition to $\xi$, the universal mixture over environments), and consumes action-data generated by maximizing the Self-AIXI policy i.e. $\arg\max_a Q_\xi^\zeta(h_t, a)$. In summary our contributions are:

- We define Self-AIXI, a novel universal agent based on self-prediction.
- We prove that Self-AIXI's Q-values converge to AIXI's Q-values asymptotically.
- We show how Self-AIXI leads to maximum Legg-Hutter [21] intelligence and inherits the self-optimization [22] property.

These results provide compelling evidence that self-prediction can effectively serve as a robust alternative to traditional planning methods.

## 2 Background

### 2.1 General Reinforcement Learning

Let $\mathcal{A}, \mathcal{O}, \mathcal{R}$ denote the (finite) action, observation and reward set respectively. Let the set of percepts be defined as $\mathcal{E} := \mathcal{O} \times \mathcal{R}$. $\Delta \mathcal{X}$ denotes a distribution over $\mathcal{X}$. Define the set of histories $\mathcal{H} := (\mathcal{A} \times \mathcal{E})^*$. A policy $\pi : \mathcal{H} \to \Delta \mathcal{A}$ is a distribution over actions given a history. An environment $\mu : \mathcal{H} \times \mathcal{A} \to \Delta \mathcal{E}$ is a distribution over percepts given history and action. We use $h_{<t} := a_1 e_1 \ldots a_{t-1} e_{t-1} = a_1 o_1 r_1 \ldots a_{t-1} o_{t-1} r_{t-1}$ to denote the history of interactions up to time $t - 1$. Given an environment $\mu$ and policy $\pi$ we use $\mu^\pi$ to denote the induced probability measure on histories, i.e. $h_{<t}$ is assumed to be sampled from $\mu^\pi$. We are interested in how well agents (policies) perform. We measure this performance by the future expected sum of discounted rewards, called value function.

**Definition 1** (Value function and optimal policy). *The value $V_\nu^\pi : \mathcal{H} \to [0, 1]$ of a policy $\pi$ in an environment $\nu$, and discount factor $\gamma$ given a history $h_{<t} \in \mathcal{H}$ is defined as*

$$V_\nu^\pi(h_{<t}) := \frac{1}{\Gamma_t} \mathbb{E}_\nu^\pi \left[ \sum_{k=t}^\infty \gamma^{k-t} r_k \,\middle|\, h_{<t} \right]$$

*Where $\Gamma_t = \sum_{k=t}^\infty \gamma^{k-t} = \frac{1}{1-\gamma}$ is the normalization factor of the discount. The optimal value is defined as $V_\nu^*(h_{<t}) := \sup_\pi V_\nu^\pi(h_{<t})$, the set of optimal policies with respect to that value is defined as $\arg\max_\pi V_\nu^\pi(h_{<t})$, and an optimal policy with respect to the value is defined as $\pi_\nu^*(\cdot|h_{<t}) \in \arg\max_\pi V_\nu^\pi(h_{<t})$.*

We can similarly define the action value or Q-value function as $Q_\nu^\pi(h_{<t}, a_t) := V_\nu^\pi(h_{<t} a_t)$, which allows to write the pseudo-recursive Bellman forms

$$V_\nu^\pi(h_{<t}) = \sum_{a_t \in \mathcal{A}} \pi(a_t|h_{<t}) Q_\nu^\pi(h_{<t}, a_t), \quad Q_\nu^\pi(h_{<t}, a_t) = \sum_{e_t \in \mathcal{E}} \nu(e_t|h_{<t} a_t) \left( r_t + \gamma V_\nu^\pi(h_{1:t}) \right) \quad (1)$$

We use throughout the paper the Total-Variation distance as a convenient way to compare two policies.

**Definition 2** (TV distance). *We define the Total Variation (TV) distance of two measures $\nu_1^{\pi_1}, \nu_2^{\pi_2}$ between timesteps $t$ and $t \leq m \leq \infty$ as*

$$D_m(\nu_1^{\pi_1}, \nu_2^{\pi_2}|h_{<t}) := \sup_{H \subseteq (\mathcal{A} \times \mathcal{E})^m} |\nu_1^{\pi_1}(H|h_{<t}) - \nu_2^{\pi_2}(H|h_{<t})| \quad \leq \quad 1$$

The following lemma, useful for our later proofs, shows that TV distance is an upper bound for value function difference in general reinforcement learning.

**Lemma 3** ([23, Lemma 4.17]). *For any two policies $\pi_1, \pi_2$ and two environments $\nu_1, \nu_2$*

$$|V_{\nu_1}^{\pi_1}(h_{<t}) - V_{\nu_2}^{\pi_2}(h_{<t})| \leq D_{\infty}(\nu_1^{\pi_1}, \nu_2^{\pi_2}|h_{<t})$$

## 2.2   Universal Artificial Intelligence

We achieve universality by considering a Bayesian mixture over a class of potential environments $\mathcal{M}$ which is large enough so as to include the true environment the agent is interacting with. A typical choice is the class of all computable functions.

**Definition 4** (Bayesian mixture environment). *The Bayesian mixture over the class of environments $\mathcal{M}$ is defined as*

$$\xi(e_t|h_{<t}a_t) := \sum_{\nu \in \mathcal{M}} w(\nu|h_{<t})\nu(e_t|h_{<t}a_t) \quad \text{with posterior} \quad w(\nu|h_{1:t}) := w(\nu|h_{<t})\frac{\nu(e_t|h_{<t}a_t)}{\xi(e_t|h_{<t}a_t)}$$

*and $w(\nu|\epsilon) := w(\nu)$ is the prior probability of $\nu$.*

Given this environment mixture $\xi$, we define AI$\xi$ (AIXI) as acting optimally with respect to $\xi$.

**Definition 5** (AI$\xi$). *AIXI, the optimal Bayesian agent is defined as*

$$\pi_{\xi}^*(h_{<t}) := \arg\max_a Q_{\xi}^*(h_{<t}, a)$$

The following results prove that the AIXI agent achieves maximal intelligence in the Legg-Hutter sense and is self-optimizing. These are important to have in mind throughout the paper since we will prove the same type of results for our Self-AIXI agent.

Legg-Hutter Intelligence measures the performance of an agent $\pi$ in a wide range of environments $\mathcal{M}$, weighted by their p(oste)rior plausibility $w(\cdot)$:

**Definition 6** (LH Intelligence [21]). *The Legg-Hutter Intelligence measure $\Upsilon$ of a policy $\pi$ given a history $h_{<t}$ is defined as*

$$\Upsilon(\pi|h_{<t}) := \sum_{\nu \in \mathcal{M}} w(\nu|h_{<t})V_{\nu}^{\pi}(h_{<t}) = V_{\xi}^{\pi}(h_{<t})$$

It has been shown that if $\mathcal{M}$ is such that there exists a policy (sequence) which is able to achieve strong asymptotic optimality then AIXI can achieve this as well. This property is known as self-optimizing.

**Theorem 7** (AIXI is Self-optimizing [14]). *Let $\mu$ be some environment. If there is a policy $\pi$ and a sequence of policies $\overline{\pi_1}, \overline{\pi_2}\ldots$ such that for all $\nu \in \mathcal{M}$*

$$V_{\nu}^*(h_{<t}) - V_{\nu}^{\overline{\pi_t}}(h_{<t}) \to 0 \quad as \quad t \to \infty \quad \mu^{\pi}\text{-almost surely} \tag{2}$$

*then*

$$V_{\nu}^*(h_{<t}) - V_{\nu}^{\pi_{\xi}^*}(h_{<t}) \to 0 \quad as \quad t \to \infty \quad \mu^{\pi}\text{-almost surely}$$

*If $\pi = \pi_{\xi}^*$ and Equation 2 holds for all $\mu \in \mathcal{M}$, then $\pi_{\xi}^*$ is strongly asymptotically optimal in the class $\mathcal{M}$.*

For a taxonomy of classes that do (not) allow for self-optimizing policies, see [21, Figure 3.1]. A challenge that is difficult to overcome is that for geometric discounting, there may not yet be widely-recognized classes of environments that allow self-optimizing policies.

# 3 Self-prediction in General Reinforcement Learning

We are ready to define our Self-AIXI agent that is based on *self-prediction* instead of pure planning. We start with our definition of the mixture policy $\zeta$ and discuss the choices of model class and prior.

**Definition 8** (Bayesian mixture policy). *The Bayesian mixture over the class of policies $\mathcal{P}$ is defined as*

$$\zeta(a_t|h_{<t}) := \sum_{\pi \in \mathcal{P}} \omega(\pi|h_{<t})\pi(a_t|h_{<t}) \quad \text{with posterior} \quad \omega(\pi|h_{1:t}) := \omega(\pi|h_{<t})\frac{\pi(a_t|h_{<t})}{\zeta(a_t|h_{<t})}$$

*and $\omega(\pi|\epsilon) := \omega(\pi)$ is the prior probability of $\pi$. Note $\omega(\cdot)$ is a prior for the mixture policy, while $w(\cdot)$ was a prior for the mixture environment.*

**Lemma 9** (Linearity of Q-values). *For all $\mu$, $\pi$, $h_{<t}$ and $a_t$ we have*

$$Q_\xi^\pi(h_{<t}, a_t) = \sum_{\nu \in \mathcal{M}} w(\nu|h_{<t})Q_\nu^\pi(h_{<t}, a_t), \qquad Q_\mu^\zeta(h_{<t}, a_t) = \sum_{\pi \in \mathcal{P}} \omega(\pi|h_{<t})Q_\mu^\pi(h_{<t}, a_t)$$

From the linearity of $Q$-values we can define the following $Q$-value over the Bayesian mixture environment and Bayesian mixture policy.

$$Q_\xi^\zeta(h_{<t}, a_t) := \sum_{\pi \in \mathcal{P}} \omega(\pi|h_{<t}) \sum_{\nu \in \mathcal{M}} w(\nu|h_{<t})Q_\nu^\pi(h_{<t}, a_t) \tag{3}$$

**Definition 10** (Self-AIXI). *The agent Self-AIXI is defined as taking the (one step) optimal action with respect to $Q_\xi^\zeta$. Formally,*

$$\pi_S(h_{<t}) := \arg\max_{a_t} Q_\xi^\zeta(h_{<t}, a_t)$$

**Remark 11** (Self-AIXI does not optimize the future.). *Importantly, Self-AIXI maximizes the Q-values from the mixture policy instead of the optimal policy, which means that there is no need to optimize the future but it needs to know the Q-values of the current $\zeta$-mixture. Pragmatically, these Q-values are on-policy, typically easier to estimate than the optimal off-policy values.*

Note how action selection and history interact with the policy-mixture. The action selected by the Self-AIXI agent necessarily improves, by definition, over the current value estimates $\max_{a_t} Q_\xi^\zeta(h_{<t}, a_t) \geq V_\xi^\zeta(h_{<t})$. Then, this action is added to the next history $h_{<t+1}$ which is consumed by the policy-mixture at the next time step i.e., $\zeta(a_{t+1}|h_{<t+1})$. This leads to the following remark.

**Remark 12** (Self-predicting minimal improvements.). *The policy-mixture does Bayesian inference over the incoming self-generated action-data and makes better predictions over time. Thus, Self-AIXI is self-predicting its own small improvements made by the $\arg\max$ operation in Definition 10. In the case of using the largest class of policies (all computable functions), the mixture is a Solomonoff predictor with enough power to represent, within itself, the policy evaluation and improvement operation.*

Showing that the self-prediction process employed by Self-AIXI converges to AIXI is the aim of our theoretical results in Section 4.

## 3.1 Choice of class and prior

When choosing an environment class $\mathcal{M}$ we need to pick one large enough so as to contain the true environment $\mu$; the larger the class the weaker the assumption that $\mu \in \mathcal{M}$. However, the larger the class we choose the harder it will be to compute the Bayesian mixture of that class, to the point that $\xi$ may become incomputable (or at least more incomputable than any $\nu \in \mathcal{M}$). With all of this in mind there are two directions that this can be taken: The first is to choose the largest class such that computing $\xi$ is still tractable and the second is to choose the largest possible class such that it contains all possible environments our universe could contain (or be). In the first case we can choose something like the class of variable-order MDPs with efficient Context Tree Weighting

(CTW) algorithm [24] for $\xi$, as is done in [20]. In the second case we can pick the set of all (cumulatively lower-) semicomputable semimeasures [25]. Without going too much into the choice of prior, a simplicity-based prior is an ideal choice. There is an interesting poly-time computable class of all measures of logarithmic complexity [26] for which $\xi \in \mathcal{M}$ (($\alpha, \gamma$)-simple measures). One easy way of achieving $\xi \in \mathcal{M}$ is to add $\xi$ to $\mathcal{M}$ with some weight $w_0' > 0$ and renormalize $w_i \rightsquigarrow w_i' = (1 - w_0')w_i$, then $\xi' \equiv \xi \in \mathcal{M}' := \{\xi\} \cup \mathcal{M}$. We can apply the same logic to the policy class $\mathcal{P}$ and policy prior $\omega(\pi)$ as we just did for $\mathcal{M}$ and $w(\mu)$, similarly we need to have that $\pi_S \in \mathcal{P}$. We leave the full investigation of policy classes to future work. Full discussion on the choice of class $\mathcal{M}$ and prior $w(\nu)$ are beyond the scope of this work [14, 27].

## 3.2 Experimental Results

While our work is mainly theoretical, we also conducted experiments (see Appendix B) comparing self-prediction, using a Self-AIXI approximation, against the pure planning approach, using an AIXI approximation, using Context Tree Weighting as predictor and Monte-Carlo Tree Search for the Q-value estimates. In short, the Self-AIXI approximation outperforms the AIXI approximation in three environments and performs equally in the remaining two.

# 4 Theoretical Results

In this section we will demonstrate the intelligence of Self-AIXI by showing that it asymptotically converges to AIXI in expectation. We specifically chose this criterion over other types of optimality, since Cesaro and Almost-Sure asymptotic optimality (converging to the optimal policy) are too strong [28–30], and are not satisfied by AIXI. In fact, agents which satisfy these also die with certainty [31].

The goal is to show that Self-AIXI eventually performs as well as the most intelligent agent, and AIXI is the most intelligent agent [21]. We will do this by proving an asymptotic convergence result to the Bayes optimal policy in all environments.

To summarize the results of this section:

- Theorem 18: $\forall \mu. \ V_\xi^{\pi_S} \to V_\xi^*$ in $\mu^{\pi_S}$-expectation.

- Theorem 21: $\forall \mu. \ V_\mu^{\pi_S} \to V_\mu^{\pi_\xi^*}$ in $\mu^{\pi_S}$-expectation.

- Theorem 22: $V_\mu^{\pi_S} \to V_\mu^* \ \mu^{\pi_S}$-almost surely, under similar conditions to the AIXI Self-optimizing result from Theorem 7.

The initial three results successively broaden in scope, with the second being a more general form of the first, and the third further generalizing the second, holding for all $\mu$ instead of just $\xi$. All proofs not found in this section can be found in the appendix.

## 4.1 Expected AIXI-like behavior

To start with, we will present a useful Lemma about the environment mixture $\xi$:

**Lemma 13** (Convergence of $\xi$ to $\mu$ in TV). *For any environment $\mu \in \mathcal{M}$ and policy $\pi$ we have*

$$D_\infty(\xi^\pi, \mu^\pi | h_{<t}) \to 0 \quad as \quad t \to \infty \quad \mu^\pi\text{-almost surely}$$

*Proof.* Dominance $\xi^\pi(\cdot) \geq w(\mu)\mu^\pi(\cdot)$ implies that $\mu^\pi$ is absolutely continuous with respect to $\xi^\pi$ and therefore $\xi^\pi$ merges strongly with $\mu^\pi$ [32]. For full details and definitions of strong merging and absolute continuity see [23]. □

We can derive a dual to Lemma 13 and restate it in terms of asymptotic convergence in expectation:

**Lemma 14** (Convergence of $\zeta$ to $\pi$ in expectation). *If $\pi \in \mathcal{P}$ then for all $\nu$ we have*

$$\mathbb{E}_\nu^\pi[D_\infty(\nu^\pi, \nu^\zeta | h_{<t})] \to 0 \quad as \quad t \to \infty$$

*Proof.* The proof is dual to the proof of Lemma 13 with policies and environments exchanged. Finally, almost sure convergence implies convergence in expectation for bounded random variables. □

Now we can show that if a policy is close to one-step optimal, then it is also close to optimal (this result is independent of $\xi$ and will work for any environment, but we are only interested in the $\xi$ case).

**Lemma 15** (A one-step good policy is close to optimal). *Let* $\Delta(h) := |V_\xi^*(h) - V_\xi^\pi(h)|$ *with* $h \in (\mathcal{A} \times \mathcal{E})^t$ *for* $t \geq t_0 \in \mathbb{N}$.

$$\text{If} \quad \mathbb{E}_\xi^\pi | \max_a Q_\xi^\pi(h,a) - V_\xi^\pi(h)| < \beta \quad \forall t \geq t_0$$

$$\text{and} \quad \mathbb{E}_\xi^\pi[\max_a \sum_e \xi(e|ha)\Delta(hae)] \leq (1+\alpha)\mathbb{E}_\xi^\pi\Delta(hae) \quad \forall t \geq t_0$$

$$\text{then} \quad \mathbb{E}_\xi^\pi\Delta(h) < \frac{\beta}{1-\gamma(1+\alpha)} \quad \forall t \geq t_0 \quad \text{provided} \quad 1+\alpha < 1/\gamma$$

Using the previous Lemma we can show our main result about the behavior of Self-AIXI. This result shows that the performance of Self-AIXI converges to that of AIXI on (expected) histories generated by Self-AIXI. We call a policy that satisfied the second condition for some $1 + \alpha < 1/\gamma$ for some (large) $t_0$, *sensible off-policy*. We conjecture this plausible condition holds for $\pi_S$, bar some exotic counter-example classes $\mathcal{M}$.

**Theorem 16** (Self-AIXI converges to AIXI in $\xi$-expectation). *Assuming $\pi_S$ is sensible off-policy,*

$$\mathbb{E}_\xi^{\pi_S} \left[ V_\xi^*(h_{<t}) - V_\xi^{\pi_S}(h_{<t}) \right] \to 0 \quad as \quad t \to \infty.$$

*Proof.*
$$0 \overset{(a)}{\leq} \max_a Q_\xi^\zeta(h,a) - V_\xi^\zeta(h) \overset{(b)}{=} V_\xi^{\pi_S}(h) - V_\xi^\zeta(h) \overset{(c)}{\leq} D_\infty(\xi^{\pi_S}, \xi^\zeta|h)$$

where (a) and (b) follow from

$$V_\xi^{\pi_S}(h) \equiv \max_a Q_\xi^\zeta(h,a) \geq \sum_a \zeta(a|h)Q_\xi^\zeta(h,a) = V_\xi^\zeta(h)$$

and (c) follows from Lemma 3. Let $\beta > 0$ and $h \in (\mathcal{A} \times \mathcal{E})^t$. Now Lemma 14 and the above implies that there exists a $t_0$ such that for all $t \geq t_0$,

$$0 \leq \mathbb{E}_\xi^{\pi_S}[\max_a Q_\xi^\zeta(h,a) - V_\xi^\zeta(h)] \leq \mathbb{E}_\xi^{\pi_S} D_\infty(\xi^{\pi_S}, \xi^\zeta|h) < \beta$$

Taking the expectation of $0 \leq V_\xi^* - V_\xi^{\pi_S} \leq V_\xi^* - V_\xi^\zeta$ (which follows from $V_\xi^* \geq V_\xi^{\pi_S} \geq V_\xi^\zeta$), Lemma 15 implies

$$0 \leq \mathbb{E}_\xi^{\pi_S}[V_\xi^*(h) - V_\xi^{\pi_S}(h)] \leq \mathbb{E}_\xi^{\pi_S}[V_\xi^* - V_\xi^\zeta] \leq \beta/(1 - \gamma(1+\alpha)).$$

Since $\beta > 0$ was arbitrary, we get the $\mathbb{E}_\xi^{\pi_S}[V_\xi^*(h) - V_\xi^{\pi_S}(h)] \to 0$. $\qquad\square$

The previous theorem also holds in expectation over the true environment $\mu$, instead of in expectation w.r.t. $\xi$, but first we need a small lemma:

**Lemma 17** ($\mathbb{E}_\xi^\pi \to 0$ implies $\mathbb{E}_\mu^\pi \to 0$). *If $\pi$ is such that*

$$\mathbb{E}_\xi^\pi \left[ V_\xi^*(h_{<t}) - V_\xi^\pi(h_{<t}) \right] \to 0 \quad as \quad t \to \infty.$$

*then for all $\mu \in \mathcal{M}$ we have*

$$\mathbb{E}_\mu^\pi \left[ V_\xi^*(h_{<t}) - V_\xi^\pi(h_{<t}) \right] \to 0 \quad as \quad t \to \infty.$$

We can now present the main Theorem of the paper, this theorem states that as Self-AIXI interacts with the true environment $\mu$ Self-AIXI will learn to act like the optimal Bayesian agent AIXI.

**Theorem 18** (Main Theorem: Self-AIXI converges to AIXI in $\mu$-expectation). *For all $\mu \in \mathcal{M}$ we have*

$$\mathbb{E}_\mu^{\pi_S} \left[ V_\xi^*(h_{<t}) - V_\xi^{\pi_S}(h_{<t}) \right] \to 0 \quad as \quad t \to \infty.$$

*Proof.* Theorem 16 and Lemma 17. $\qquad\square$

We can rewrite the above theorem in the form of the Universal Intelligence measure $\Upsilon$,

**Corollary 19** (LH Form of Theorem 18)**.** *For all $\mu \in \mathcal{M}$*

$$\mathbb{E}_\mu^{\pi_S} \left[ \max_\pi \Upsilon(\pi|h_{<t}) - \Upsilon(\pi_S|h_{<t}) \right] \to 0 \quad as \quad t \to \infty$$

Finally we can generalise our previous result to work for the value function of any $\mu$. This is important because we are most interested in the performance of our agent on the true environment. We first need to show that convergence of the value function with $\xi$ implies convergence of the value function with $\mu$ (in $\mu$-expectation).

**Lemma 20** ($V_\xi^{\pi'} \to V_\xi^\pi$ implies $V_\mu^{\pi'} \to V_\mu^\pi$ in $\mu$-expectation)**.** *If $\pi$ is such that for all $\mu \in \mathcal{M}$*

$$\mathbb{E}_\mu^\pi \left[ V_\xi^{\pi'}(h_{<t}) - V_\xi^\pi(h_{<t}) \right] \to 0 \quad as \quad t \to \infty.$$

*and $D_\infty(\mu^{\pi'}, \xi^{\pi'}|h_{<t}) \to 0 \; \mu^\pi$-almost surely then we have*

$$\mathbb{E}_\mu^\pi \left[ V_\mu^{\pi'}(h_{<t}) - V_\mu^\pi(h_{<t}) \right] \to 0 \quad as \quad t \to \infty.$$

Now we can present a convergence in value functions with the true environment $\mu$.

**Theorem 21** (Self-AIXI converges to the $\xi$-optimal agent in $\mu$-expectation)**.** *For all $\mu \in \mathcal{M}$ if $D_\infty(\mu^{\pi_\xi^*}, \xi^{\pi_\xi^*}|h_{<t}) \to 0 \; \mu^{\pi_S}$-almost surely then we have*

$$\mathbb{E}_\mu^{\pi_S} \left[ |V_\mu^{\pi_\xi^*}(h_{<t}) - V_\mu^{\pi_S}(h_{<t})| \right] \to 0 \quad as \quad t \to \infty.$$

*Proof.* Immediate result form Theorem 18 and Lemma 20 $\qquad\qquad\qquad\qquad\square$

This result states that as Self-AIXI interacts with the true environment $\mu$, the expected future performance difference between Self-AIXI and AIXI in $\mu$ will go to zero, that is to say, they will (asymptotically) have equal performance in $\mu$.

## 4.2 Self-optimization

We have previously shown the performance of Self-AIXI using convergence of value functions $\mu$-expectation. We can also demonstrate the performance of Self-AIXI in a more direct way, using self-optimization. That is, showing that if it is possible for a policy to learn to be optimal in all environments in a given class, then Self-AIXI will learn to be optimal in all environments in this class.

Self-AIXI can self-optimize in the same way as AIXI, with an additional assumption on the self model. The additional assumption is that for all $t, h_{<t}$ we have $V_\xi^\zeta(h_{<t}) \geq V_\xi^{\overline{\pi}_t}(h_{<t}) - \epsilon_t$.

**Theorem 22** (Self-AIXI is Self-optimizing)**.** *Let $\mu$ be some environment. If there is a policy $\pi$ and a sequence of policies $\overline{\pi}_1, \overline{\pi}_2 \ldots$ all contained within $\mathcal{P}$ such that for all $t, h_{<t}$ we have $V_\xi^\zeta(h_{<t}) \geq V_\xi^{\overline{\pi}_t}(h_{<t}) - \epsilon_t$ with $\epsilon_t \to 0$, and for all $\nu \in \mathcal{M}$*

$$V_\nu^*(h_{<t}) - V_\nu^{\overline{\pi}_t}(h_{<t}) \to 0 \quad as \quad t \to \infty \quad \mu^\pi\text{-almost surely} \tag{4}$$

*then*

$$V_\nu^*(h_{<t}) - V_\nu^{\pi_S}(h_{<t}) \to 0 \quad as \quad t \to \infty \quad \mu^\pi\text{-almost surely}$$

*If $\pi = \pi_S$ and Equation 4 holds for all $\mu \in \mathcal{M}$, then $\pi_S$ is strongly asymptotically optimal in the class $\mathcal{M}$.*

There are known important cases of model classes which admit self-optimising policies such as Ergodic MDPs [33] and Ergodic k-MDPs [34]. However, we have not yet shown there exist (interesting) classes $\mathcal{M}$ and $\mathcal{P}$ where the assumption that for all $t, h_{<t}$ we have $V_\xi^\zeta(h_{<t}) \geq V_\xi^{\overline{\pi}_t}(h_{<t}) - \epsilon_t$ with $\epsilon_t \to 0$ and the other self-optimizing assumptions hold. We leave this to future work.

| | Policy Learning (or distillation) | Planning | Environment Model | Bayes Optimal | Practical |
|---|---|---|---|---|---|
| AIXI | No | Yes | Yes | Yes | No |
| Self-AIXI | Yes | No | Yes | Yes | No |
| MC-AIXI-CTW | No | Yes | Yes | Yes | Yes |
| $\mu$-Zero | Yes | Yes | Yes | No | Yes |
| DQN | Yes | No | No | No | Yes |
| CnC | No | No | No | Yes | Yes |
| PhiMDP | No | No | No | No | No |

Table 1: Comparison between different methods

All models in the table learn an (action) value function.

## 5 Related work

### 5.1 Universal Artificial Intelligence

There has been much work studying the Bayesian optimal agent AIXI and extending it in various ways [14, 21, 23]. Some of these extensions include: Knowledge seeking agent (KSA), an extension of AIXI which seeks information leading to a form of maximal exploration [35, 36]. BayesExp, which combines the exploiting of AIXI and exploring of KSA in a way that allows the agent to be weakly asymptotically optimal [37, 38]. An AIXI variant that uses Thompson sampling [39] for additional exploration is asymptotically optimal in expectation [28]. Inquisitive Agent (Inq) uses an information gaining strategy based on expeditions to achieve strong asymptotic optimality [40].

AIXI achieves (by definition) Bayesian optimality, and although there are problems with this notion of optimality, it is widely agreed to be the (current) best choice for an optimality criterion. Every notion of optimality possesses inherit difficulties; these difficulties have been extensively studied Leike and Hutter [29], Lattimore and Hutter [30], Cohen et al. [31].

### 5.2 Comparison with other methods in the literature

To overcome the computational burden of the AIXI's planning phase, approximation techniques usually relax the planning problem into an estimation problem [41, 42]. The main approach for off-the-shelf estimation methods such as Monte Carlo Tree Search (MCTS) is an (optimistic) estimation of the Q-values. Though seemingly different, planning and learning, in the general case, can be seen as following very similar computational processes as shown by [43]. Other practical methods which reduce or remove the planning problem are distributional reinforcement learning [44], Compress and Control (CnC) [45] and feature reinforcement learning (FRL) [46, 47]. In distributional reinforcement learning and CnC powerful sequence prediction methods are utilized to estimate the distribution of returns directly. While it does not remove the planning entirely, FRL adaptively compresses the general RL problem to a more simple (often MDP) problem and solves that problem by traditional methods. The planning required in the simple problem is often far less than in the general problem.

In Table 1 we compare some exemplary agents where each column denotes whether they use some form of learning (or policy distillation), whether they use explicit search/planning, whether they learn an explicit model of the environment, if they are a Bayes-optimal agent, and if there are practical in the sense that there is an actual implementation of the agent. As can be seen, Self-AIXI is the only Bayes-optimal agent that is universal and exploits learning instead of planning.

### 5.3 Agents modeling themselves

Self-prediction can be thought as an agent modeling itself. Within the universal AI literature, there are other works that go further and drop the dualism of traditional reinforcement learning for the (more realistic) physicalism where agents are contained within their own environment. In this case many complications arise which previous work tries to address. For example, in [48] the behavior of several agents able to modify their own source code is analyzed. In [49], Orseau et. al. presents a space-time embedded definition of the value function and intelligence, and in [50] Leike et. al. aims to solve the grain of truth (a self-referential) problem of Bayesian agents in GRL, showing that there

is a non-trivial class of environments for which the optimal Bayesian agent over that class is within the class.

## 5.4 Self Prediction in Traditional RL

Recently traditional reinforcement learning has begun to embrace self-prediction. In [51] Predictions of Bootstrapped Latents (PBL) is presented as a method which can use predictive representations on multitask environments. [52] used Self-Predictive Representations (SPR) to train agents to anticipate future latent states, which showed improvements on Atari games benchmarks. In [53], exploration in visually complex environments was streamlined with BYOL-Explore using a bootstrapped prediction. Additionally, [54] underscores the challenges of representation collapse in self-predictive learning and introduces bidirectional learning to address this. Together, these studies underscore the evolving landscape of self-predictive strategies in traditional RL.

# 6 Discussion, Future Work and Limitations

**Practical considerations for computing the mixture component in each step.**    Computing the Self-AIXI policy requires estimating Q-values using the Bayesian mixture policy $\zeta(a_t|h_{<t})$. Naively, this means evaluating each policy in the mixture and computing its respective posterior weight. However, more efficient schemes can be derived, e.g., similar to how the CTW algorithm computes the Bayesian mixture without explicitly evaluating each hypothesis and computing the (marginalizing) sum over hypotheses in each step.

**Leveraging large sequence models.**    Powerful sequence predictors like large language models (LLMs) are the cornerstone of modern general AI due to their impressive zero- and few-shot performance [55–57]. An open question in AGI research is how to leverage the strength of sequence prediction not only for environment modeling, but also to derive goal-directed policies; how to build agents from predictors. We have shown how to shift computational effort for planning (in AIXI) can be transferred into effort devoted to prediction in Self-AIXI, making self-prediction theoretically very well suited to exploit strong predictors such as LLMs. In contrast to similar ad-hoc approaches [58], Self-AIXI is derived from sound theoretical design principles and comes with theoretical guarantees.

**Safety implications.**    Self-AIXI constitutes a theoretical blueprint for building powerful general AI agents that may be very suitable to leverage strong predictors, such as LLMs, in practice. The implications of such agents have recently been discussed in the philosophical and technical literature, and concerns regarding their safety have been raised [59, 60]. Self-AIXI comes with some interesting favorable safety properties. First and foremost, Self-AIXI is derived from sound reasoning and decision-making principles: Bayesian posterior inference and rational optimization of rewards. Self-AIXI comes with a well-understood theoretical model to study abstract properties, guarantees and bounds. An advantage over AIXI, by design, is the self-model of the agent, which can be analyzed on its own. The explicit separation of beliefs over the environment and posterior over the own policy leads to an agent that is more interpretable and can be verified more easily compared to a monolithic agent where components are mixed. If the self-predictor used is not a Bayesian one, but instead some black box predictor, then both the guarantees of Self-AIXI and the guarantees on interpretability are no longer maintained. We strongly encourage an open and differentiated wider discussion on the safety of (universal) RL agents, and believe that theoretical models, like Self-AIXI, can facilitate this discussion by providing a formal and mathematically concrete formulation.

**Limitations.**    Our theoretical results require that the Self-AIXI policy is part of the policy class under consideration, i.e., $\pi_S \in \mathcal{P}$ (meaning that the mixture policy is itself part of the policy class). Generally, if there is a policy in the policy class that is "close" to the optimal policy (in terms of minimal KL divergence), Self-AIXI will still perform well. This result can be stated formally using known formulations and bounds for Bayesian mixtures. We leave a detailed mathematical analysis for future work, since the aim of this work is to introduce the theory of Self-AIXI. Another issue is that Self-AIXI, like the original AIXI, is incomputable for environment classes required for (strict) universality. To tackle this, many practical approximations of AIXI have been proposed in the literature, which provide straightforward starting points for similar approximations for Self-AIXI.

Designing and studying non-trivial practical approximations of Self-AIXI is beyond the scope of this paper.

**Future work.** On the theoretical side there are several promising directions of future work: exploring which non-trivial policy classes $\mathcal{P}$ satisfy $\pi_S \in \mathcal{P}$ (or replacing this assumption); extending this work with a sampling-first approach (e.g. Thompson Sampling). On the practical side the first step is to empirically demonstrate the performance of the self-prediction approach proposed in this paper. We believe that especially when coupled with powerful modern predictors, e.g., transformer-based sequence predictors, high-fidelity approximations of $\xi$ and $\zeta$ are possible, which can lead to strong, general agents.

# 7    Conclusion

In this paper we presented Self-AIXI, a theoretical framework for universal AI that extends AIXI by self-prediction (the agent performs inference over its own policy). The result is an agent that matches AIXIs performance guarantees, while shifting the computational focus on prediction instead of planning. This makes Self-AIXI interesting for building general agents by leveraging recent breakthroughs in sequence predictors, and combining them with RL in a sound fashion. Current approaches to do this, such as RL from human/artificial feedback (RLHF/RLAF), are quite limited (in terms of generality of the RL formulation) and do not come with optimality or universality guarantees. We believe our results to be an important cornerstone for building agents that go beyond zero- or few-shot transfer, towards rapid and data efficient RL.

**Acknowledgments**

We thank Tor Lattimore, Laurent Orseau and Anian Ruoss for their helpful feedback and insightful discussions.

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

# A Proofs

**Lemma 15** (A one-step good policy is close to optimal)**.** *Let* $\Delta(h) := |V_\xi^*(h) - V_\xi^\pi(h)|$ *with* $h \in (\mathcal{A} \times \mathcal{E})^t$ *for* $t \geq t_0 \in \mathbb{N}$.

$$\text{If}\quad \mathbb{E}_\xi^\pi | \max_a Q_\xi^\pi(h,a) - V_\xi^\pi(h)| < \beta \quad \forall t \geq t_0$$

$$\text{and}\quad \mathbb{E}_\xi^\pi[\max_a \textstyle\sum_e \xi(e|ha)\Delta(hae)] \leq (1+\alpha)\mathbb{E}_\xi^\pi \Delta(hae) \quad \forall t \geq t_0$$

$$\text{then}\quad \mathbb{E}_\xi^\pi \Delta(h) < \tfrac{\beta}{1-\gamma(1+\alpha)} \quad \forall t \geq t_0 \quad \text{provided}\quad 1+\alpha < 1/\gamma$$

*Proof.* Let $\delta := \sup_{t \geq t_0} \mathbb{E}\Delta(h)$, where $h \in (\mathcal{A} \times \mathcal{E})^t$ and $\mathbb{E}$ is short for $\mathbb{E}_\xi^\pi$.

$$
\begin{aligned}
\mathbb{E}\Delta(h) &= \big| \max_a Q_\xi^*(h,a) - V_\xi^\pi(h) \big| \\
&= \mathbb{E} \big| \max_a Q_\xi^*(h,a) - \max_a Q_\xi^\pi(h,a) + \max_a Q_\xi^\pi(h,a) - V_\xi^\pi(h) \big| \\
&\leq \mathbb{E} \big| \max_a Q_\xi^\pi(h,a) - V_\xi^\pi(h) \big| + \mathbb{E} \big| \max_a Q_\xi^*(h,a) - \max_a Q_\xi^\pi(h,a) \big| \\
&\overset{(1)}{<} \beta + \mathbb{E} \big| \max_a \sum_e \xi(e|ha)\big(r + \gamma V_\xi^*(hae)\big) - \max_a \sum_e \xi(e|ha)\big(r + \gamma V_\xi^\pi(hae)\big) \big| \\
&\leq \beta + \gamma \mathbb{E} \max_a \sum_e \xi(e|ha)|V_\xi^*(hae) - V_\xi^\pi(hae)| \\
&\leq \beta + \gamma(1+\alpha)\mathbb{E}\Delta(hae)
\end{aligned}
$$

Taking $\sup_{t \geq t_0}$ on both sides implies $\delta < \beta + \gamma(1+\alpha)\delta$ implies $\delta < \beta/(1-\gamma(1+\alpha))$. $\quad\square\quad\square$

**Lemma 17** ($\mathbb{E}_\xi^\pi \to 0$ implies $\mathbb{E}_\mu^\pi \to 0$)**.** *If $\pi$ is such that*

$$\mathbb{E}_\xi^\pi \big[ V_\xi^*(h_{<t}) - V_\xi^\pi(h_{<t}) \big] \to 0 \quad \text{as}\quad t \to \infty.$$

*then for all $\mu \in \mathcal{M}$ we have*

$$\mathbb{E}_\mu^\pi \big[ V_\xi^*(h_{<t}) - V_\xi^\pi(h_{<t}) \big] \to 0 \quad \text{as}\quad t \to \infty.$$

*Proof.*

$$\mathbb{E}_\mu^\pi \big[ V_\xi^*(h_{<t}) - V_\xi^\pi(h_{<t}) \big] \leq \frac{1}{w(\mu)}\mathbb{E}_\xi^\pi \big[ V_\xi^*(h_{<t}) - V_\xi^\pi(h_{<t}) \big] \to 0$$

by the dominance of $\xi(\cdot) \geq w(\mu)\mu(\cdot)$. $\quad\square$

**Lemma 20** ($V_\xi^{\pi'} \to V_\xi^\pi$ implies $V_\mu^{\pi'} \to V_\mu^\pi$ in $\mu$-expectation)**.** *If $\pi$ is such that for all $\mu \in \mathcal{M}$*

$$\mathbb{E}_\mu^\pi \big[ V_\xi^{\pi'}(h_{<t}) - V_\xi^\pi(h_{<t}) \big] \to 0 \quad \text{as}\quad t \to \infty.$$

*and $D_\infty(\mu^{\pi'}, \xi^{\pi'}|h_{<t}) \to 0$ $\mu^\pi$-almost surely then we have*

$$\mathbb{E}_\mu^\pi \big[ V_\mu^{\pi'}(h_{<t}) - V_\mu^\pi(h_{<t}) \big] \to 0 \quad \text{as}\quad t \to \infty.$$

*Proof.*

$$
\begin{aligned}
&\mathbb{E}_\mu^\pi \Big[ |V_\mu^{\pi'}(h_{<t}) - V_\mu^\pi(h_{<t})| \Big] \\
&= \mathbb{E}_\mu^\pi \Big[ |V_\mu^{\pi'}(h_{<t}) - V_\xi^{\pi'}(h_{<t}) + V_\xi^{\pi'}(h_{<t}) - V_\xi^\pi(h_{<t}) + V_\xi^\pi(h_{<t}) - V_\mu^\pi(h_{<t})| \Big] \\
&\leq \mathbb{E}_\mu^\pi \Big[ |V_\mu^{\pi'}(h_{<t}) - V_\xi^{\pi'}(h_{<t})| \Big] + \mathbb{E}_\mu^\pi \Big[ |V_\xi^{\pi'}(h_{<t}) - V_\xi^\pi(h_{<t})| \Big] + \mathbb{E}_\mu^\pi \Big[ |V_\xi^\pi(h_{<t}) - V_\mu^\pi(h_{<t})| \Big]
\end{aligned}
$$

The second and third term go to 0 as $t \to \infty$ by the assumptions and Lemma 3 with Lemma 13. The first term goes to 0 as $D_\infty(\mu^{\pi'}, \xi^{\pi'}|h_{<t}) \to 0$ $\mu^\pi$-almost surely implies $\mathbb{E}_\mu^\pi \Big[ D_\infty(\mu^{\pi'}, \xi^{\pi'}|h_{<t}) \Big] \to 0$ and we have $\mathbb{E}_\mu^\pi \Big[ |V_\mu^{\pi'}(h_{<t}) - V_\xi^{\pi'}(h_{<t})| \Big] \leq \mathbb{E}_\mu^\pi \Big[ D_\infty(\mu^{\pi'}, \xi^{\pi'}|h_{<t}) \Big]$.

$\quad\square$

**Theorem 22** (Self-AIXI is Self-optimizing). *Let $\mu$ be some environment. If there is a policy $\pi$ and a sequence of policies $\overline{\pi}_1, \overline{\pi}_2 \ldots$ all contained within $\mathcal{P}$ such that for all $t, h_{<t}$ we have $V_\xi^\zeta(h_{<t}) \geq V_\xi^{\overline{\pi}_t}(h_{<t}) - \epsilon_t$ with $\epsilon_t \to 0$, and for all $\nu \in \mathcal{M}$*

$$V_\nu^*(h_{<t}) - V_\nu^{\overline{\pi}_t}(h_{<t}) \to 0 \quad as \quad t \to \infty \quad \mu^\pi\text{-almost surely} \tag{4}$$

*then*

$$V_\nu^*(h_{<t}) - V_\nu^{\pi_S}(h_{<t}) \to 0 \quad as \quad t \to \infty \quad \mu^\pi\text{-almost surely}$$

*If $\pi = \pi_S$ and Equation 4 holds for all $\mu \in \mathcal{M}$, then $\pi_S$ is strongly asymptotically optimal in the class $\mathcal{M}$.*

*Proof.*

$$0 \leq w(\mu|h_{<t}) \left( V_\mu^*(h_{<t}) - V_\mu^{\pi_S}(h_{<t}) \right) \tag{5}$$

$$\leq \sum_{\nu \in \mathcal{M}} w(\nu|h_{<t}) \left( V_\nu^*(h_{<t}) - V_\nu^{\pi_S}(h_{<t}) \right) \tag{6}$$

$$= \sum_{\nu \in \mathcal{M}} w(\nu|h_{<t}) V_\nu^*(h_{<t}) - V_\xi^{\pi_S}(h_{<t}) \tag{7}$$

$$\leq \sum_{\nu \in \mathcal{M}} w(\nu|h_{<t}) V_\nu^*(h_{<t}) - V_\xi^\zeta(h_{<t}) \tag{8}$$

$$\leq \sum_{\nu \in \mathcal{M}} w(\nu|h_{<t}) V_\nu^*(h_{<t}) - V_\xi^{\overline{\pi}_t}(h_{<t}) + \epsilon_t \tag{9}$$

$$= \sum_{\nu \in \mathcal{M}} w(\nu|h_{<t}) \left( V_\nu^*(h_{<t}) - V_\nu^{\overline{\pi}_t}(h_{<t}) \right) + \epsilon_t \tag{10}$$

$$\to 0 \tag{11}$$

Equation 6 comes from adding positive terms. Equations 7 and 10 comes from the linearity of the value function. Equation 8 comes from $\pi_S$ being one step optimal then following $\zeta$ and . Equation 9 comes from the assumptions. Lastly, 11 comes from Equation 4 and [14, Lem.5.28ii].

$w(\mu|h_{<t}) \not\to 0$ as $h_{<t}$ is generated from $\mu^\pi$ (for more details see Self-Optimizing proof in [14]). Therefore $V_\mu^*(h_{<t}) - V_\mu^{\pi_S}(h_{<t}) \to 0$ $\mu^\pi$-almost surely.

$\square$

# B Experimental Validation

In this section we heuristically evaluate the effect of self-modelling using the AIXI approximation provided by Veness et al. This AIXI approximation uses Monte-Carlo Tree Search (MCTS) [61] and the Context Tree Weighting [62] technique respectively to approximate the expectimax planning and Bayesian environment learning components of the AIXI optimal agent. At each timestep the approximation performs a MCTS to estimate the action-value function with respect to its CTW-based environment model. We consider two agent configurations: MC-AIXI, which chooses each action uniformly at random in the rollout stage of MCTS, i.e. $\hat{\pi}^*_{\hat{\xi},u}(a_t \mid h_{<t}) := 1/|\mathcal{A}|$, and matches identically the original configuration in [20]; and MC-Self-AIXI, which uses Context Tree Weighting, i.e. $\hat{\pi}^*_{\hat{\xi},\hat{\zeta}}(a_t \mid h_{<t}) := \mathrm{CTW}_d(a_t \mid h_{<t})$, to predict the outcome of each MCTS decision.

**Experimental Setup.** For all experiments, we use a finite $m$-horizon undiscounted return setup, with each domain specific horizon choice given by Table 3 in [20]. The CTW depth parameter to *both* the environment model ($\hat{\xi} := \mathrm{CTW}_d$) and the self-prediction model ($\hat{\zeta} := \mathrm{CTW}_d$) was also chosen to match Table 3 in [20]. We evaluated across 5 stochastic, partially observable and history dependent domains: Cheeze Maze, Kuhn Poker, 4x4 Grid, Tiger and Biased Scissor/Paper/Rock. The description for each of these domains can once again be found in [20]. Each environment was evaluated by performing a single online run across $10^4$ timesteps; here a smaller number of timesteps was used than in [20] so that we could better assess the transient effects on performance improvement. At each timestep $t$, both agents pick either a random action with probability $\epsilon_t := 0.2 \times 0.999^t$, or otherwise return the estimated best action according to action-value estimates computed with 500 simulations.

**Results.** Figure 1 shows learning curves for the Cheeze Maze, Tiger and 4x4 Grid domains respectively. The vertical axis shows the average reward per timestep obtained over the previous 2000 time steps. In these 3 domains we see that self-modelling gives a qualitative advantage over the original MC-AIXI agent given the restricted data and computational budget described above, approaching optimal performance in each domain; note that optimal performance can be obtained too with the original MC-AIXI agent, albeit with more compute and data, as described in [20]. This matches the theory which states that both agents will converge to optimal performance, but shows that self-modelling can, in some cases, converge to the optimal behavior more quickly. In two of the easier domains, ScissorPaperRock and Kuhn Poker, there was no significant difference in performance between the two methods. The final performance (as evaluated by the average reward per step over the final 2000 timesteps) of each agent configuration is shown in Table 2.

| Agent | Kuhn Poker | Biased Scissor Paper Rock | Cheese Maze | Tiger | 4×4 Grid |
|---|---|---|---|---|---|
| MC-AIXI | 1.98 | 0.17 | -1.0 | -0.29 | 0.0 |
| MC-Self-AIXI | 1.93 | 0.22 | 1.08 | 0.5 | 0.24 |

Table 2: Final performance of MC-AIXI-CTW and MC-Self-AIXI-CTW after $10^4$ time steps.

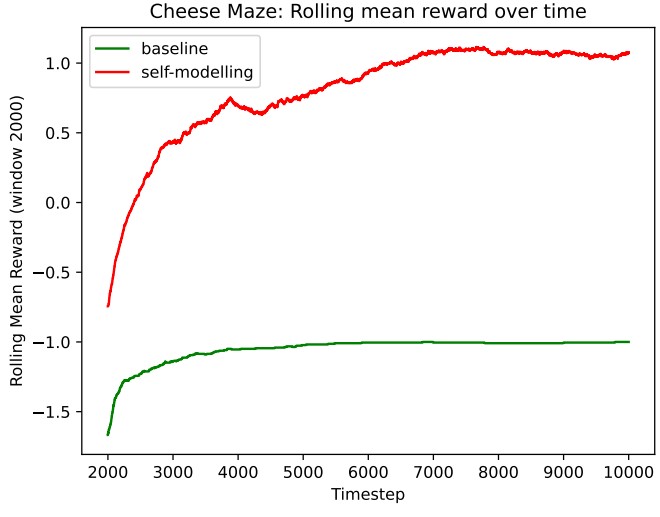

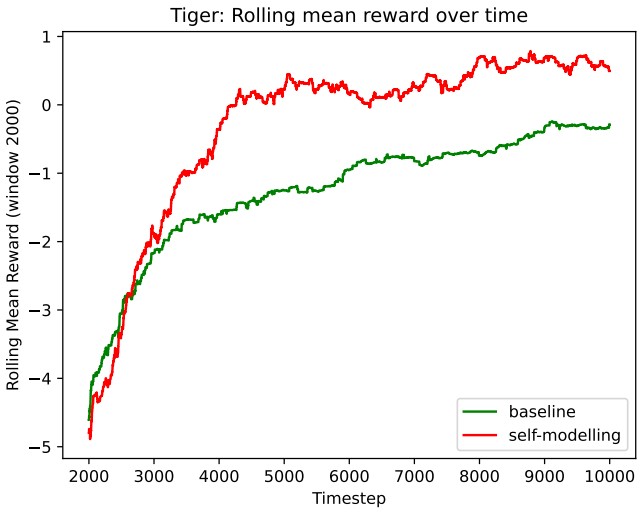

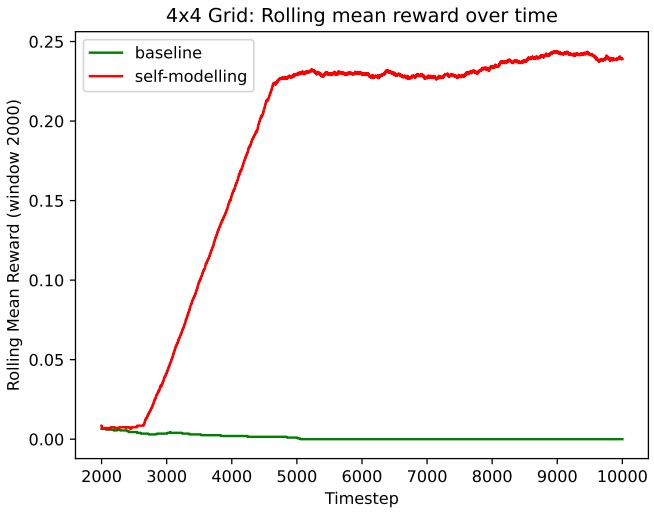

Figure 1: Rolling average reward per step for MC-AIXI (green) and MC-Self-AIXI (red).

