# OpenReview forum: "Self-Predictive Universal AI"
_NeurIPS.cc/2023/Conference — NeurIPS 2023 poster_

### Official Review · Reviewer_xEuK · 2023-07-07

**Soundness:** 3 good
**Presentation:** 3 good
**Contribution:** 1 poor
**Rating:** 5
**Confidence:** 2

**Summary:**

The authors propose Self-AIXI, an extension of the universal Bayes-optimal agent (AIXI), combining it with self-prediction for use in combination with reinforcement learning instead of planning. They provide theoretical evidence that Self-AIXI inherits optimality properties from AIXI.

**Strengths:**

- Overall, the paper is well-written and theoretically sound.
- All claims are supported by extensive theoretical analysis.

**Weaknesses:**

- As Self-AIXI inherits its optimality properties from AIXI, the overall contriubution seems limited.
- Even though the proposed extension seems interesting, I fail to find proper motivation or clear comunication of the sigificance of the contribution by the authors.

**Questions:**

- I failed to find any motivation for increasing the complexity of AIXI even more by incorporating reinforcement learning instead of planning. Could you elaborate on that?

Further comments / suggestions:

- The related work presented in 5 seems to have little relation to Self-AIXI and could be extended.
- Table 1 could be explained more extensively.
- Regarding the safety implications mentioned in 6, i tend to disagree, as the addition of further (possilby blackbox) models mostly hinders interpretability, thus safety-critical validation.

**Limitations:**

Limitations are clearly pointed out.

---

> ### Author Rebuttal · Authors · 2023-08-09
>
> We thank the reviewer for their comments.
> * “As Self-AIXI inherits its optimality properties from AIXI, the overall contribution seems limited.”: We argue that one of the key results (and the main effort) of our paper is to show that Self-AIXI can perform equally optimally and achieve AIXI optimality. Once we can show that “Self-AIXI” converges to “AIXI” (in the sense spelled out in the paper), this does allow us to “inherit” many known optimality properties of AIXI without having to prove these properties again for Self-AIXI. Constructing an algorithm that improves over AIXI’s optimality properties would indeed be a significant contribution to the field of (universal) AI, but unless there are substantial flaws in the optimality proofs of AIXI this is impossible.
> * “I fail to find proper motivation or clear communication of the significance of the contribution by the authors.”: We have added this motivation and statement of significance to the introduction and main contributions of our paper.
> * “I failed to find any motivation for increasing the complexity of AIXI even more by incorporating reinforcement learning instead of planning. Could you elaborate on that?”: In this work we have demonstrated an alternate formula for an optimal agent. The complexity that is added from self prediction is taken away from additional planning. While we do have some added complexity in terms of the self model, since we are already modeling the environment the added complexity will be just our model complexity, and we will save not having to do the traditional planning.
> * “The related work presented in 5 seems to have little relation to Self-AIXI and could be extended.”: We would like to kindly ask whether the reviewer has any particular references in mind?
> * “Table 1 could be explained more extensively.”: We have expanded our explanation of Table 1.
> * “Regarding the safety implications mentioned in 6, I tend to disagree, as the addition of further (possilbly blackbox) models mostly hinders interpretability, thus safety-critical validation.”: We have expanded our safety discussion by this point raised by the reviewer.

---

> > ### Comment · Reviewer_xEuK · 2023-08-18
> > **Thank you for your rebuttal.**
> >
> > Thank you for the clarification regarding the contribution of your work and your remarks on the changed complextiy of the approach.
> > Regarding the related work presented, further references to work on the connecetion of self-prediction and RL would be helpful to classify your approach.
> > Nevertheless I will take your remarks into consideration and update my review accordingly.

---

### Official Review · Reviewer_dcXk · 2023-07-07

**Soundness:** 3 good
**Presentation:** 3 good
**Contribution:** 3 good
**Rating:** 6
**Confidence:** 3

**Summary:**

This paper presents a new universal Bayesian AI agent framework that uses reinforcement learning to predict its own learning to form incrementally better and ultimately optimal policies. The paper then shows theoretically that this agent, Self-AIXI, converges to the same policies as a planning based universal, Bayesoptimal agent, AIXI, thus showing that the learning based system itself will converge to optimal. The main contribution of the paper is a theoretical analysis of the self-predicting agent in the context of universal, Bayes-optimal agent frameworks.

**Strengths:**

The paper presents a self-predicting agent that uses reinforcement learning to form a universal AI agent that learns an optimal policy in the Bayes sense over a distribution of environments. The formal analysis and proofs seem sound and the final result is of interest to persons in universal AI as it introduces a differnt approach to Bayes optimal agents that could further down lead to more practical agents in this class. While (as acknowledged by the authors) self-AIXI right now is not a practical framework and the assumptions in the self-optimization proof are very strong, the paper provides a point and potential direction and stepping stone towards more practical universal AI.

**Weaknesses:**

The main weaknesses of the paper are that it is mainly a formal, theoretical endeavor with no direct known (or anticipated) practical applications. This is clearly acknowledged by the authors in terms of the practical application as well as there not being any practical classes of environments for which the assumptions of the central self-optimization proof would hold.

**Questions:**

My main question would be whether the authors see any direct  way in which the presented system would be practically applyable ? Outside the perview of universal AI, would there be subproblem domains for which this agent would be practical ? And how small would those have to be ? (Having such - even small - domains could make the paper of interest to a larger part of the community).

**Limitations:**

The authors very clearly state the main limitations of the preented work, which lie in the practicality of the approach and in the strong assumption they had to make for the self-optimization proof (and in particular that there currently are no practical environment distributions for which these assumptions hold).

---

> ### Author Rebuttal · Authors · 2023-08-09
>
> We thank the reviewer for their comments.
> * “mainly a formal, theoretical endeavor with no direct known (or anticipated) practical applications”: As the reviewer correctly states, the main goal of this paper is to work out the theory for self-prediction and distillation in the limit. We can anticipate practical applications, as is demonstrated by the experiment described in the general response.
> * “My main question would be whether the authors see any direct way in which the presented system would be practically applyable ?”: We are conducting a proof-of-concept experiment (see general response) to determine if using a self predictor can improve performance of an AIXI approximation.

---

### Official Review · Reviewer_AePK · 2023-07-16

**Soundness:** 3 good
**Presentation:** 3 good
**Contribution:** 2 fair
**Rating:** 6
**Confidence:** 1

**Summary:**

The paper introduces a reinforcement learning version of an agent
(Self-Predictive Universal AI == Self-AIXI) that converges to the
AIXI universal Bayes optimal agent. The advanage of the approach is
it is based on learning/reinforcement learning rather than planning and
thus opens up new avenues for practical approximations to the AIXI scheme.
The term 'self' is based on learning a more compact (distilled) model
for predicting the agent's actions.
A good property-based comparison with other theoretical and practical implementations is given in table 1.


**Strengths:**

The claim is based on a mathematical proof. I could not follow all of the details,
but the general structure of the argument and proof seemed sound.


**Weaknesses:**

This is a theoretical existence proof rather than an demonstation by implementation,
and thus it is unclear whether there are benefits of the approach, eg: faster learner for
a given error rate, more accurate practical approximation to AIXI, etc. The paper
states that both AIXI and Self-AIXI in their pure form are computationally intractable.


**Questions:**

It would be helpful if the theorems/lemmas/etc had a plain-English statement to help
present the concepts clearly as well as precisely. A table of symbols would also help.


**Limitations:**

Somewhat - the paper recognizes that the approach is still computtionally intractable.

---

> ### Author Rebuttal · Authors · 2023-08-09
>
> We thank the reviewer for their comments.
> * “it is unclear whether there are benefits of the approach, eg: faster learner for a given error rate, more accurate practical approximation to AIXI, etc”: From a theoretical perspective the benefits are that we have shown how predictors can be used to assist the planning effort of the agent. We are conducting a proof-of-concept experiment (see general response) to determine if the self prediction of Self-AIXI can be used to improve an AIXI approximation.
> * “It would be helpful if the theorems/lemmas/etc had a plain-English statement to help present the concepts clearly as well as precisely. A table of symbols would also help.”: We have added plain-English statements of our main theorems and lemmas to the paper.

---

### Official Review · Reviewer_sXfu · 2023-07-26

**Soundness:** 2 fair
**Presentation:** 3 good
**Contribution:** 3 good
**Rating:** 5
**Confidence:** 2

**Summary:**

The paper proposes a Bayesian agent for reinforcement learning called Self-AIXI. Self-AIXI learns from its own Q-value maximizing actions and performs action prediction, instead of using extensive search to find optimal action at each step like AIXI. The paper proves that Self-AIXI converges to AIXI and therefore shares optimality properties of AIXI.

**Strengths:**

A proof of optimality of policy distillation methods: by proving the optimality of Self-AIXI, a theoretical guarantee is given to policy distillation methods, which could enable further analysis of the theoretical properties of such approach and help design practical methods with a guarantee.

Extending the theory of general reinforcement learning to include learning: the policy learning component is introduced into the theoretical framework of RL agents, which could potentially enable analysis of strong RL agents based on extensive learning.

Theory framework is clear and easy to follow: the definition of Self-AIXI and the proof framework is presented neatly and very easy to follow. The proofs are straightforward and do not require advanced techniques.

**Weaknesses:**

A central condition of the main theorem might be too strong or hard to evaluate: the main theorem requires that $\pi_S$ be a sensible off-policy that satisfies conditions of Lemma 15, but the authors leave the proof of the satisfiability as a conjecture. This would make it hard to evaluate the significance of the condition and consequently the significance of the main theorem.

Connection between empirical discussion and theory work is kind of ambiguous: the authors try to motivate the  theory of Self-AIXI from empirical work on policy distillation, and terms such as "learning", "distillation" and "self-prediction" are used throughout the paper to bridge theoretical discussions and empirical practice. However, the connection between empirical claims such as "method X uses learning" and theoretical results of Self-AIXI seems ambiguous without a clear definition of such terms in mathematical language. It would make the connection much clearer if  "learning" and "self-prediction" could be formally defined within the framework of reinforcement learning.

*Regarding originality and contribution over prior work, I would like to leave the assessment to other reviewers as I am not familiar with the related work.*

**Questions:**

I am not particularly familiar with reinforcement learning theory, but from a generalized perspective, I found the following points in the paper a little bit confusing. Hopefully, these questions might be helpful if the authors want to address a broader audience.

* Self-prediction is defined in the paper as the process of predicting the action generated by the agent itself, but it does not seem intuitively clear where such prediction happens in the definition of Self-AIXI.

* The authors also regard self-prediction as a form of learning, but it is not very explicit where the learning component is.

* If one wants to optimize a parameterized policy, what would be the objective function? If the objective is to predict the action with max Q-value, why can AIXI not employ learning to predict its optimal action selected from planning?

Other questions:

* What does $\gamma$ represent in Lemma 15? It does not seem very clear from the context.

**Limitations:**

The authors adequately addressed the limitations of the theory of Self-AIXI in their paper.

---

> ### Author Rebuttal · Authors · 2023-08-09
>
> We thank the reviewer for their comments.
> * “the main theorem requires that $\pi_S$ be a sensible off-policy that satisfies conditions of Lemma 15, but the authors leave the proof of the satisfiability as a conjecture. This would make it hard to evaluate the significance of the condition and consequently the significance of the main theorem.”: In response to the concerns about the satisfiability of $\pi_S$, it's important to note that its satisfiability is intrinsically linked to the chosen model and policy class. While we acknowledge the importance of a comprehensive investigation into the categorization and conditions under which this can be achieved, such an exhaustive exploration is beyond the purview of this paper.
> * Connection between theory and empirical algorithms too ambiguous (“learning” and “self-prediction” would need a formal definition): we have added formal definitions for “learning” and self-prediction to the paper.
>
> **Questions:**
> * “Self-prediction is defined in the paper as the process of predicting the action generated by the agent itself, but it does not seem intuitively clear where such prediction happens in the definition of Self-AIXI.”: This prediction lies in the mixture zeta, this mixture uses the past actions in the history to model and predict the future actions of the agent.
> * “The authors also regard self-prediction as a form of learning, but it is not very explicit where the learning component is.”: The learning done in the self prediction is Bayesian learning as we use a Bayesian mixture as the self predictor.
> * “If one wants to optimize a parameterized policy, what would be the objective function? If the objective is to predict the action with max Q-value, why can AIXI not employ learning to predict its optimal action selected from planning?”: The goal of the agent is to achieve the maximal expected reward, defined by the value function. AIXI will take the optimal action from planning, AIXI is the optimal agent. Self-AIXI shows us that when implementing agents, we can use powerful predictors to assist (or almost completely remove) the planning component
> * “What does $\gamma$ represent in Lemma 15? It does not seem very clear from the context.”: $\gamma$ is the discount factor (see Definition 1).

---

### Official Review · Reviewer_mKJS · 2023-07-31

**Soundness:** 2 fair
**Presentation:** 3 good
**Contribution:** 2 fair
**Rating:** 5
**Confidence:** 3

**Summary:**

This paper focuses on the issue of inefficiency in planning for AIXI agent, particularly lacking an alternative universal agent that maximally exploits learning and distillation. To address the issue, this work proposed a new method, named self-AIXI, that maximally exploits self-prediction instead of planning by doing exact Bayesian inference on the policy space. Further, theoretical results were provided to prove that self-AIXI's Q-values can converge to AIXI's Q-values asymptotically.


**Strengths:**

Strengths:
- It's an interesting work for self-AIXI to consider both planning and learning to improve the efficiency and guarantee the optimal solution and its convergence to the gold-standard AIXI agent. Besides, this work also provides theoretical analysis for the optimality condition, compared to the existing empirically successful model-based reinforcement learning framework like MuZero.
- The paper is well-written and I can read it smoothly.

**Weaknesses:**

Weaknesses:
- This is a theoretical paper, but only introducing the theory of self-AIXI might be a little bit less. It would be better if some preliminary results towards a concrete algorithm can be included in this work. This is because there exists some practical work for AIXI and model-based RL like MuZero (target to theoretical or/and empirical problems), so it's not quite clear why this self-AIXI is really needed, although there were some discussions shown in the paper. I do appreciate this work, so I'd like to hear more about this from the authors.


**Questions:**

- See my comments in the weakness.

---

> ### Author Rebuttal · Authors · 2023-08-09
>
> We thank the reviewer for their comments.
> * “It would be better if some preliminary results towards a concrete algorithm can be included in this work.”: We are conducting some proof-of-concept experiments with MC-AIXI-CTW, which will be included in the updated manuscript. Please see our main response for details. Additionally we have added more discussion to the paper towards bridging the gap between theory and practical algorithms.
> * “it's not quite clear why this self-AIXI is really needed”: We see Self-AIXI as an important step in the path to understanding the optimal behaviour of general agents. This theory shows us that optimal behaviour can be achieved with minimal planning, if a sufficiently powerful predictor is used.

---

### Author Rebuttal · Authors · 2023-08-09

We thank the reviewers for their helpful comments. We are pleased to see that all reviewers found our paper easy to read (giving the maximum score of 3 for presentation), and all reviewers consider the theoretical claims and analysis sound and easy to follow. We respond to comments shared by reviewers in this general response and also provide a detailed response to each reviewer individually under their reviews.

The main criticism, shared by reviewers, is that it is difficult to connect the theory to concrete implementations and applications. We understand this criticism, which applies to most theoretical work in universal AI, and want to emphasize that the main goal of this manuscript is to provide a sound theoretical construction and analysis of an alternative to the planning-centric approach, enabling in theory the development of more efficient and practical universal AI agents. We strongly agree with dcXk: “the paper provides a point and potential direction and stepping stone towards more practical universal AI.”. Closing the gap towards practical implementations is the next critical step in this line of research, but we believe that this requires quite a bit of setup and clarification that deserves a full separate publication. To show that the practical impact of the theory is not completely beyond reach, we are conducting some proof-of-concept experiments with Monte Carlo AIXI Context Tree Weighting (MC-AIXI-CTW)[1]. In these experiments we are comparing a Self-AIXI approximation with an AIXI approximation. The AIXI approximation is the base MC-AIXI-CTW, and the Self-AIXI approximation is MC-AIXI-CTW equipped with a self predictor (CTW predicting its own actions used within the MC rollout). This comparison is done on the environments in [1].

We want to thank the reviewers for pointing out parts of the manuscript that can be improved, in particular the clarification of Lemma 15 (sXfu), the formal definitions of “learning” and “self-prediction” (sXfu),motivation for incorporating reinforcement learning (xEuK), and plain-English statements of the main theorems and lemmas (AePK). We have also added a discussion on how to bridge the gap towards practical applications (dcXk) and concrete algorithms (mKJS).

[1] Veness, J., Ng, K. S., Hutter, M., Uther, W., & Silver, D. (2011). A Monte-Carlo AIXI approximation. Journal of Artificial Intelligence Research, 40, 95-142.

---

### Decision · Program_Chairs · 2023-09-21

**Decision:**

Accept (poster)

**Comment:**

This paper introduces a new 'universal' Bayesian AI agent framework that uses RL to predict its own learning and incrementally obtain better—and ultimately optimal—policies. In particular, the paper introduces a new method, called self-AIXI, that maximally exploits self-prediction instead of planning by performing exact Bayesian inference on the policy space. The authors prove that this approach converges to the same policies as universal planning-based, Bayes-optimal agents (AIXI), thereby showing that their novel learning-based system will also converge to optimal solutions. This paper's main contribution is a theoretical analysis of the self-predicting agent in the context of universal, Bayes-optimal frameworks.

The reviewers agree that taking into account both planning and learning to improve efficiency (while guaranteeing optimality and convergence to the gold-standard AIXI solution) is interesting. Furthermore, reviewers also acknowledged the importance of this work's theoretical analyses regarding the optimality condition, compared to existing—and empirically successful—model-based RL frameworks such as MuZero.

A few concerns were raised. First, many reviewers—although aware of the correctness of the paper's theoretical results—argued that this work's practical significance remains vague and unclear. They understand that this is a theoretical paper but argued that by only introducing the theory of self-AIXI, it is challenging for readers to assess its relevance to the field. During the discussion, it was brought up that it would have been better if preliminary results towards a concrete algorithm could have been included. After all, there exists practical work (which target both theoretical and empirical problem) for AIXI and model-based RL like MuZero, and so it is not entirely clear why/when/whether self-AIXI is needed or advantageous compared to existing approaches. Another reviewer was concerned with a central condition of the main theorem, which they believe might be too strong or hard to evaluate. In particular, the main theorem requires that $\pi_S$ satisfy the conditions of Lemma 15, but the authors leave the satisfiability proof as a conjecture. This makes it difficult to evaluate the significance of this condition and, consequently, the significance of the main theorem itself. Finally, many reviewers commented on how adding plain English explanations of the theorems would help clarify many points discussed in this paper and clarify its contributions. As it is, there was a consensus that readers may likely have trouble following the claims and results presented in this work and, consequently, properly evaluating its relevance.

Having said that, the reviewers acknowledged this work's contributions while emphasizing the points that need improvement before publication. They encouraged the authors to explore these points and address all limitations that were brought up in the reviews when preparing an updated version of this paper.